# SARS-CoV-2 Infection and Oral Health: Therapeutic Opportunities and Challenges

**DOI:** 10.3390/jcm10010156

**Published:** 2021-01-05

**Authors:** Christopher J. Coke, Brandon Davison, Niariah Fields, Jared Fletcher, Joseph Rollings, Leilani Roberson, Kishore B. Challagundla, Chethan Sampath, James Cade, Cherae Farmer-Dixon, Pandu R. Gangula

**Affiliations:** 1Department of Oral Diagnostic Sciences & Research, School of Dentistry, Meharry Medical College, Nashville, TN 37208, USA; ccoke20@email.mmc.edu (C.J.C.); bdavison19@email.mmc.edu (B.D.); nfields19@email.mmc.edu (N.F.); jfletcher19@email.mmc.edu (J.F.); jrollings15@email.mmc.edu (J.R.); lroberson20@email.mmc.edu (L.R.); csampath@mmc.edu (C.S.); jcade@mmc.edu (J.C.); cdixon@mmc.edu (C.F.-D.); 2Department of Biochemistry & Molecular Biology, The Fred and Pamela Buffet Cancer Center, University of Nebraska Medical Center, Omaha, NE 68198, USA; kishore.challagundla@unmc.edu; 3The Children’s Health Research Institute, University of Nebraska Medical Center, Omaha, NE 68198, USA

**Keywords:** COVID-19, periodontitis, Angiotensin Converting Enzyme 2 (ACE-2), saliva, inflammation, oxidative stress, dental practice

## Abstract

The novel corona virus, Severe Acute Respiratory Syndrome Coronavirus-2 (SARS-CoV-2), and the disease it causes, COVID-19 (Coronavirus Disease-2019) have had multi-faceted effects on a number of lives on a global scale both directly and indirectly. A growing body of evidence suggest that COVID-19 patients experience several oral health problems such as dry mouth, mucosal blistering, mouth rash, lip necrosis, and loss of taste and smell. Periodontal disease (PD), a severe inflammatory gum disease, may worsen the symptoms associated with COVID-19. Routine dental and periodontal treatment may help decrease the symptoms of COVID-19. PD is more prevalent among patients experiencing metabolic diseases such as obesity, diabetes mellitus and cardiovascular risk. Studies have shown that these patients are highly susceptible for SARS-CoV-2 infection. Pro-inflammatory cytokines and oxidative stress known to contribute to the development of PD and other metabolic diseases are highly elevated among COVID-19 patients. Periodontal health may help to determine the severity of COVID-19 infection. Accumulating evidence shows that African-Americans (AAs) and vulnerable populations are disproportionately susceptible to PD, metabolic diseases and COVID-19 compared to other ethnicities in the United States. Dentistry and dental healthcare professionals are particularly susceptible to this virus due to the transferability via the oral cavity and the use of aerosol creating instruments that are ubiquitous in this field. In this review, we attempt to provide a comprehensive and updated source of information about SARS-CoV-2/COVID-19 and the various effects it has had on the dental profession and patients visits to dental clinics. Finally, this review is a valuable resource for the management of oral hygiene and reduction of the severity of infection.

## 1. Introduction

Corona viruses are a diversified class of viruses with zoonotic origin, highly transmitted in humans, causing mild to severe respiratory infections. In 2002 and 2012, respectively, two highly pathogenic coronaviruses emerging in humans were (a) severe acute respiratory syndrome coronavirus (SARS-CoV) and (b) Middle East respiratory syndrome coronavirus (MERS-CoV), causing deadly respiratory illness. At the end of 2019, a novel coronavirus designated as SARS-CoV-2 emerged as a pneumonia of the lower respiratory tract in a patient in Wuhan, China on December 29, 2019 [1,2]. The World Health Organization (WHO) classified COVID-19, the disease associated with the virus SARS-CoV-2, as a global pandemic. Several patients with pneumonia were then reported to have contracted the novel virus, linked to a Hunan South Province China Seafood Market in Wuhan, Hubei Province, China [3]. This virus has currently spread to approximately 215 countries with over forty nine million cases and over 1.24 million deaths worldwide [4].

SARS-CoV-2 differs from SARS-CoV due to its higher level of transmissibility and pandemic risk. SARS-CoV-2 has a greater significant reproductive number (R), the statistic used to determine how infectious the agent is, at 2.9. SARS-CoV had an R of (1.77) [2,5,6]. It is this specific trait of SARS-CoV-2 that makes it more of a global concern than SARS-CoV. SARS-CoV-2, like SARS-CoV, is transmitted via aerosols and can pass from human to human [7]. The incubation period for the virus is from 1–14 days and the infected patient can remain contagious even through its latency period. Once symptoms are observed, a positive diagnosis is achieved by performing real-time PCR (RT-PCR) to positively detect SARS-CoV-2 RNA in various bodily fluids, including sputum, throat swabs, and secretions of the lower respiratory tract and from fecal and blood samples [2,5]. Alternative detection methods using serological/antibody testing are also employed but there are conflicting conclusions about these methods’ effectiveness. Due to the recent discovery of this virus, its full effects on the body are not yet totally understood. However, scientists and dentists believe the oral cavity may play a crucial role in the early diagnosis and treatment of this disease [8,9].

The pneumonia-like symptoms of COVID-19 include fever, cough, myalgia or fatigue, and complicated dyspnea. However, there are reported symptoms including headache, diarrhea, hemoptysis, runny nose, and phlegm-producing cough [2]. Symptoms in the most severe cases rapidly progress to acute respiratory distress syndrome, respiratory failure, multiple organ failure and death [5]. These patients often experience oral and gastrointestinal complications, loss of taste and smell [2]. Patients having underlying health complications such as diabetes, cancer, cardiovascular diseases (CVD) and hypertension are more susceptible to developing COVID-19 [2,5]. In this Review, we summarize the current understanding of the nature of SARS-CoV-2/COVID-19 and its link to oral health. Based on recently published findings, this comprehensive Review covers the epidemiology/origin, cellular pathways involved, and drug–drug interactions of SARS-CoV-2 with respect to oral health and dentist perspectives.

## 2. Epidemiological/Viral Origin Data

SARS-CoV-2 was first discovered and isolated in Wuhan, China. The virus was isolated from a patient who suffered from pneumonia-like symptoms including fever, cough, and myalgia/fatigue. Three other cases were soon found and the outbreak was linked to a local “wet market”. To confirm the infection source of SARS-Cov-2, Centers for Disease Control and Prevention (CDC) researchers collected 585 samples from the Huanan Seafood Market in Wuhan, Hubei Province, China between January 1–12, 2020. Though the original transmission is thought to be animal-to-human in nature, it is now clear that the virus has adapted a human-to-human transmission pattern. With a now recognized effective reproductive number(R) of 2.9, researchers declared SARS-CoV-2 as one of the more transmissible viruses. Other studies suggested that the basic reproduction range (R_0_) is between 2.6–4.71 with an average incubation time within the range of 2–11 days [2,5,6].

Important epidemiological factors include age, sex, race, age at death, susceptible populations, and mortality rate. To date COVID-19 affects populations regardless of age, with most cases between 35 and 55 years [2,10,11]. Susceptible to death from a COVID-19 related infection are patients 75 years and older. As of Nov 4, 2020 in the mortality rate for all 75+ years, COVID-19 patients are at 57% (Figure 1A) [12]. With age a susceptibility factor, healthcare workers and researchers have also noted that people with co-morbidities, poor immune function, long-term use of immunosuppressants, and surgery history before admission are also more susceptible to worse outcomes from a COVID-19 infection [2,13,14,15,16,17]. There are higher rates of infection in males (~59–68%) compared to females suggesting that female sex hormones may have a beneficial role in protecting against COVID-19 [2,6,14].

The mortality rate for COVID-19 is one factor that is under dispute. Between 29 December 2019–28 January 2020, the mortality rate was estimated at between 2.3–11% [2,14,18,19]. As of 1 May 2020 the Infection Fatality Rate (IFR) was 1.4%, meaning 1.4% of people infected with SARS-CoV-2 have a fatal outcome, while 98.6% recover [2,19,20]. The total number of deaths from COVID-19 in United States alone is around 231,988 as of 4 November 2020 (Figure 1A) [12]. A comprehensive review by Alcendor provided in-depth information on the factors associated with morbidity and mortality among minority populations [21]. African Americans (AAs) and Hispanics/Latinos were disproportionately impacted by COVID-19 infection when compared with non-Hispanic Whites (Figure 1B) [21,22,23]. U.S. counties such as Hancock and Randolph County, Georgia, with majority AA population are experiencing a three-fold higher infection rate and six-fold higher death rate than White counties. The death rate in AAs ranges from 40–70% due to COVID-19. Comorbidities like hypertension and diabetes, which are tied to COVID-19 complications, disproportionately affect the AA community [21]. 

However, the alarming rates at which COVID-19 is causing mortality in AAs extends beyond these comorbidities and can be attributed to decades of spatial segregation and inequitable access to testing and treatment [21]. Periodontal disease (PD) and metabolic syndrome such as obesity, diabetes and hypertension are known to be higher among this population. Therefore, it is not surprising that the morbidity and mortality rate from COVID-19 is greater in the AA population. These populations are located in poor accommodation, and have less access to health care and education with high unemployment rates. Low socioeconomic status is a risk factor for poorer health outcomes and is forcing some individuals residing in these communities out of their homes and into the workforce. Therefore, there is an unmet need to increase the access to and effectiveness of diagnostic testing interventions and provide various educational strategies by understanding the social, ethical, and behavioral implications of testing among underserved and vulnerable populations. In addition, biomarker evaluation may also help early diagnosis and identify the risk factors associated with COVID-19 [21].

## 3. Mechanism of Infection in Oral and Overall Body Health

Poor oral health may adversely influence other parts of the body. Recent studies showed that oral manifestations are commonly noticed in about 45% of COVID-19 patients [24,25,26]. Salivary glands, tonsils, and tongue are highly sensitive for SARS-CoV-2 infection [27,28,29]. The development of infection causes loss of taste, smell, and blisters on the tongue in COVID-19 patients [30,31]. It has been reported that the pathogenic microbiome found in different parts of the body such as the oral cavity, lungs and gut enhances inflammation and oxidative burden (Figure 2 and Figure 3). Studies show PD that occurs due to gram negative bacteria can aggravate COVID-19 symptoms [32,33,34]. Co-infection with the SARS-CoV-2 virus and the pathobionts of the oral cavity plays a critical role in increasing the inflammatory response and cytokine storm. Poor oral health shows a direct connection to COVID-19 infection and to a higher risk of severe illness in patients with COVID-19 [35]. In addition, the SARS-CoV-2 virus stimulates lesions on the skin, hand, foot and mouth disease which resemble those of other viral infections [35]. Further investigations need to done to determine if the virus in COVID-19 patients causes oral manifestations [36,37].

SARS-CoV 2 is classified as a β coronavirus that infects its host by five sequential steps: attachment, penetration, biosynthesis, maturation, and release, like many viruses. There are four structural proteins identified in the nucleocapsids of coronaviruses; Spike (S), membrane (M), envelope (E), and nucleocapsid (N). The Spike is a glycoprotein that protrudes from the viral surface, contributing to diversity between coronaviruses, and setting tropism. The Spike is composed of subunits S_1_ and S_2_. S_1_ binds the host cell while S_2_ acts to fuse the host cell membrane with the viral membrane [38].

It was proposed that upon binding of the Spike protein, protease cleavage occurs at the S_1_/S_2_ that triggers priming and activation [38]. Upon cleavage, the subunits remain non-covalently bound, and S_1_ assists in the stabilization of the S_2_ subunit. In contrast, cleavage at this site allows for fusion via conformation changes that were found to be irreversible [38]. The receptor binding domain in the study done by Shang et al. was found to switch between a standing-up position and a lying-down position, more binding occurring when this domain was lying down [39]. Yuki et al. proposed that many different proteases were found to have the capability of cleaving and activating the Spike, but the furin cleavage site, specifically at the S_1_/S_2_, is believed to make coronaviruses pathogenic [39]. Another protein, pro-protein convertase (PPC) found at the Spike protein site, was found not to enhance the entry of SARS-CoV2 into the cell; however, when PCC was mutated at the site, cleavage was found not to occur, thus decreasing SARS-CoV2’s ability to enter the cell. Though researchers have elucidated most of the SARS-CoV2 mechanism of infection, work continues to use what is known to develop strategies to combat the infection and disease effectively [38,39].

According to Yuki et al., Angiotensin Converting Enzyme 2 (ACE-2) was identified as one of the key targets for SARS-CoV 2, in which its expression is high among lung epithelial cells [39]. Shang et al. were amongst a group that discovered that HeLa cells (human cervical cells), Calu-3 cells (human lung epithelial cells), and MRC-5 cells (human lung fibroblast cells) were all cells that could effectively be infected by SARS-CoV-2 due to its increased affinity for hACE2, which all of these cells either exogenously or endogenously express [38]. Studies have shown that the ACE-2 receptor is a binding site for SARS-CoV2 and helps facilitate the virus’s entry into cells [40]. ACE-2 counters the activation of the Renin-Angiotensin-Aldosterone System [41]. Discussions revolve around ACE-inhibitors potentially modifying ACE-2 receptors and the effect on the virulence of COVID-19 [41,42]. Since the SARS-Coronavirus 2 disease (COVID-19) is primarily a respiratory infection, it is worth noting that ACE2 receptors are expressed on the lung alveolar epithelial cells. Lung alveolar epithelial cells were implicated as target cells for SARS-CoV 2 [41,42,43]. While ACE inhibitor use was widely examined due to its effects on ACE2 receptors, another class of antihypertensive drugs was also investigated for similar effects.

The coronavirus infection triggers endoplasmic reticulum (ER) stress responses in infected cells, associated with increased levels of reactive oxygen species (ROS) and unfolded protein response (UPR). ER stress has an important role in cardiovascular and metabolic disease, obesity and in diabetes. NRF2 (NF-E2-related factor 2) is a redox-sensitive, basic leucine zipper transcriptional factor that upregulates antioxidant gene expression by binding to the promoter region of the antioxidant response element (ARE) [44]. NRF2 controls the expression array of the detoxifying and antioxidant defense gene in multiple tissue damage during infection [44]. In addition to regulating antioxidant genes and suppressing oxidative burden, NRF2 also regulates inflammation in the pathogenesis of various disease complications including periodontitis [45]. That SARS-CoV-2 inhibits NRF2 indicates that the virus deprives the host cells of an essential cytoprotective pathway, and it will be crucial to determine how and when during the process of the viral infection this takes place, and the underlying mechanism [44]. Binding of viral protein to ACE-2 leads to virus entry. ACE-2 gene expression in oral tissues [46,47], lungs [48,49], kidney [50], stomach [51], and colon [52] has been shown to repress NRF2 [53]. The role of NRF2 in viral infections was investigated in the context of both DNA and RNA viruses [54]. In general, viruses can benefit from either activating or inhibiting NRF2 in host cells [53]. The receptor-binding domain (RBD) located in the S protein of SARS-CoV-2 interacts with the angiotensin-converting enzyme 2 (ACE-2) of host cells to allow viral entry [55]. NRF2 is the most potent antioxidant in humans and can block the AT1R axis. NRF2 plays a key role in protecting tissue destruction by excess reactive oxygen species (ROS) and suppressing inflammation occurring in periodontitis [56]. NRF2 deficiency is known to upregulate ACE-2, whereas its activator oltipraz reduces ACE2 levels, suggesting that NRF2 activation might reduce the availability of ACE-2 for SARS-CoV-2 entry into the cell [57]. The upregulation of NRF2 signaling inhibits the overproduction of IL-6, pro-inflammatory cytokines and chemokines as well as limiting the activation of NFĸB [58]. Glycogen synthase kinase 3 beta (GSK-3β) has been reported to be elevated in adipose tissue of insulin-resistant obese rodent models and in skeletal muscle of diabetic patients [59]. GSK-3β participates in the cellular response to oxidative stress, a hallmark of several nervous system disorders through its interaction with NRF2 [59]. 

Angiotensin-converting enzyme 2, which is the receptor for SARS-CoV-2, is a regulator of vascular function by modulating nitric oxide (NO) release and oxidative stress [60,61]. NO reportedly interferes with the interaction between coronavirus viral S-protein and its cognate host receptor, ACE-2 [62]. Nitric oxide-mediated S-nitro-sylation of viral cysteine proteases and host serine protease, TMPRSS2, which are both critical in viral cellular entry, appear to be nitric oxide sensitive [60,63,64]. COVID-19 patients often experience periodontal disease [65,66], and vascular [67,68] and gastrointestinal (GI) [69,70] complications, perhaps because ACE2 receptors are widely expressed among these tissues [71,72]. A hyposalivation symptom is exhibited highly in COVID-19 patients [73,74]. Hyposalivation is severe in older ages and can be linked to higher COVID-19 infection and mortality rate [74]. ACE-2 has been reported to be present in epithelial cells of the salivary gland and clinical manifestation observed in COVID-19 patients has been linked to xerostomia [75]. The expression of ACE-2 in the minor salivary glands was higher than the lungs (lung medium post-translational modifications (PTM, transcripts per kilobase of exon model per Million mapped reads) = 1.010, minor salivary gland medium PTM = 2.013), which suggests that salivary glands could be a potential target for COVID-19 [76]. SARS-CoV RNA can also be detected in saliva before lung lesions appear [77]. The positive rate of COVID-19 in patients’ saliva can reach 91.7%, and saliva samples can also cultivate the live virus [78]. This suggests that COVID-19 transmitted by asymptomatic infection may originate from infected saliva. Most importantly, SARS-CoV-2 infection may cause only GI symptoms such as nausea, vomiting and diarrhea in some of these patients [79]. Microbial symbiosis is very common with viral infection and SARS-CoV-2 RNA has been detected in feces of COVID-19 patients [80]. NRF2 and NO synthesis can be modulated by bacterial dysbiosis [81,82,83]. Our laboratory showed that Nrf2 and NO signaling play a role in maintaining vascular and gastrointestinal function in diabetic and oral infection animal models in vivo and in vitro [84,85,86]. The above data collectively suggest that infection with SARS-Cov-2 disrupts healthy microbiome and elevates inflammation and oxidative stress. This in turn modulate Nrf2 and NO signaling and may cause abnormalities in multiple organ function including respiratory, cardiovascular and gastrointestinal function in COVID-19 patients (Figure 3).

Al-Lami et al. discussed a higher rates of SARS-CoV-2 infection in adult males (~59–68%) compared to females [87]. This observation is due to the elevated levels of endogenous sex steroid hormones such as estrogen and progesterone known to play a critical role in viral defense in premenopausal women. In contrast, testosterone may be a culprit for the viral infection in males. Higher morbidity and mortality rate due to COVID-19 observed in postmenopausal women is probably due to the decrease in endogenous sex steroid hormones [87]. Sex hormones regulate multiple organ (cardiovascular, renal, GI, etc.) functions through antioxidant and anti-inflammatory properties in various disease conditions in human and rodent models [88]. The above data suggest that elevated endogenous sex hormones are more protective against SARS-CoV-2 infection in female than in male patients. In addition, the available data strongly suggest that a common mechanism of action on cytokine storm, lung injury and endothelial damage observed in most of the co-morbidities were also noticed with COVID-19 infection. Therefore, investigating the changes in these mechanisms may help to better assess the potential severity of COVID-19 infection in both sexes.

## 4. Pre-Exiting Condition Effect on COVID-19 Outcome

There is an abundance of information available regarding the effects of pre-existing medical conditions on patients’ COVID-19 infection outcomes. Of the pre-existing conditions that researchers suspect may have an impact on the outcome of patients infected with COVID-19, hypertension has been frequently mentioned. The primary association between patients with pre-existing hypertension and COVID-19 is related to the use of Angiotensin Converting Enzyme (ACE) inhibitors, a common anti-hypertensive medication [21,40,41,60,92].

In a Japanese study, it was determined that Olmesartan, an Angiotensin Receptor Blocker that is prescribed as an antihypertensive medication, resulted in a higher urinary ACE-2 receptor than individuals not taking the medication [41,93]. Individuals with pre-existing hypertension that are currently taking one of these medications and are subsequently infected with SARS-CoV2 may be more susceptible to severe complications. 

Obesity is one of the significant independent risk factor for COVID-19 infection [94]. Obesity itself promotes chronic inflammation, vitamin D deficiency, impairs immune response and causes atelectasis [95]. Hypoxemia with impaired ventilation has been associated with abdominal obesity, which increases the severity of COVID-19 infection. SARS-CoV-2 interacts with the renin–angiotensin–aldosterone system and impairs blood potassium levels, with increased susceptibility to tachyarrhythmias, possessing a potential risk of respiratory distress syndrome [95]. Therefore, COVID-19 infected obese individuals are at an additional risk of an elevated inflammatory influx and electrolyte imbalance that proves to be a potentially deadly outcome [95].

Current evidence demonstrates that patients with diabetes are more likely to experience severe symptoms and complications than patients without diabetes due to COVID-19 infection. Hyperglycemia facilitates the virus entry into the cells since ACE2 and virus both need glucose for their function [96]. Patients with poorly controlled hyperglycemia have higher pro-apoptotic factors as well as apoptosis dependent cell death in kidneys, liver, lungs, and brain [96]. Diabetic patients are prone to more severe degrees of COVID-19 infection due to their altered Renin-angiotensin-aldosterone system (RAAS) functions which facilitate viral invasion [96]. 

Besides, PD may also be a pre-existing condition that worsens COVID-19 outcomes [35,37]. Inflammation present in periodontal infections is often caused by an immune response known as a cytokine storm [89,90]. This immune response facilitates the release of cytokines locally into the gingival causing inflammation in periodontitis; however, increased cytokine levels are also observed systemically [97]. The cytokine storm was frequently identified as a cause of adverse outcomes in COVID-19 infections including Acute Respiratory Disease and Multiple Organ Failure [97,98]. Since patients with existing PD prior to SARS-CoV2 infection are likely to have elevated cytokine levels, they may be susceptible to more severe, and fatal, outcomes. To support this, lung tissues from COVID-19 patients express the pro-inflammatory cytokines that play an essential role in the development of PD [98]. Prevalence of severe periodontitis in diabetics and non-diabetics has been found to be 59.6% and 39%, respectively [99]. Another pre-existing condition pertaining to oral health is halitosis, which occurs due to an infection either in lungs, ears, nose, throat or gastrointestinal disease. COVID-19 infection is highly prevalent in subjects with halitosis [100]. The finding of Riad et al. suggest that SARS-CoV-2 affects the upper side of the tongue epithelial cells. The proposed alteration is due to the high expression of ACE 2 receptors in the dorsal part of the tongue and around the oral mucosa [101]. Evidence suggests that the mouth is a powerful source of SARS-CoV-2 infection and transmission. The presence of underlying co-morbidities synergistically affect the clinical outcomes of COVID-19 infection.

## 5. COVID-19 from a Dental Perspective 

As a profession, dentistry deals with the human oral cavity, the main route for the spread of this disease (sneezing and coughing) [102]. This puts dentists and dental offices particularly at risk of being hubs for the spread of infection, from patients to doctors, and patients to other patients. As we learn more about this infection, it is important for dentists and dental practices to update and become as familiar as possible with all aspects of this disease (Table 1). COVID-19 infection spreads mainly through droplets that remain suspended as an aerosol [36,103,104]. Dental procedures create an increased risk for infection to patients, doctors, and staff by producing aerosols and the presence of saliva. Dental practices should have procedures in place for the prevention of transmission of biological agents. “However, the procedures adopted routinely to date have not been specifically designed for the prevention of pathogens transmissible by aerosol. Therefore, there are currently no specific guidelines for the protection of dentists against SARS-CoV-2.” [26]. In addition, there are no specific procedures that are in place to prevent transmission by aerosol, so extra precautions must be taken to help prevent the spread of COVID-19 [36,103]. Fortunately, the latest statistics show that only 0.9% of dentists surveyed (*N* = 2195) had contracted COVID-19 infection. This implies that the recommended current PPE and social distancing precautions may be sufficient in dental practices to control transmission of SARS-CoV-2 [105].

In dental practice, prevention of transmission of biological agents take place by use of PPE, decontamination and sterilization procedures. The SARS-CoV-2 virus is sensitive to ultraviolet rays and heat. If exposed to temperatures of at least 56 °Celsius (132.8 °Fahrenheit) for at least 30 min it becomes inactive. Performing a telephone triage with patients to determine if they have symptoms or have come into contact with COVID-19 will allow dental providers the ability to screen patients [108]. If a patient has responses that indicate, they might have come in contact, the patient should be informed and treatment deferred unless it is an emergency case. Dental offices should adhere to social distancing in the waiting room. The number of people who have entry into operatory rooms should be minimized to individual patients, or a single adult to accompany minors. All personal items should be left in the waiting room. Allow fresh air between patients either by open windows or medical-grade air purifiers [107]. All staff should use PPE (gloves, gowns, face shields, surgical masks, FFP1,2,3 grade masks) and dispose into medical waste bins to prevent transmission by aerosol [36,103]. Temperatures of all patients, dentists, and staff are required, additionally to proper use of PPE, and disinfection. If the patient’s/staff/dentists’ temperatures are less than 100 °F and there are no COVID-19 symptoms, patients may be treated, and the dental staff and dentist may perform treatment [109]. All equipment surfaces should be protected with barrier film, cleaned with hydroalcoholic disinfectants at concentrations above 60%, and then changed after every patient. It is suggested that patients use a mouth rinse of 1% hydrogen peroxide or 1% iodopovidone for 30 s to help lower virus concentration in the mouth [106]. Providers should perform extra-oral exams over intra-oral exams when possible to prevent stimulation of coughing. Besides, dental treatment may reduce the virus burden for several hours among infected patients. Oral hygiene and mouthwashes are being looked at for their effect on reducing the viral load of COVID-19. Chlorhexidine, a common oral rinse, demonstrates substantive uses intra-orally. However, it appears not to be effective in reducing viral load. Combining chlorhexidine with ethanol at appropriate concentrations may be a useful strategy to reduce the viral load as this utilizes the effectiveness of chlorhexidine within the mouth [35,37]. 

## 6. Psychological Effects on Dental Patients and HealthCare Providers

Within the dental community, the psychological impacts of COVID-19 are vast. They affect not only dentists and patients, but also dentists’ family, and staff. As stated earlier, the nature of the profession places dentists at an increased risk of becoming exposed to COVID-19 and spreading it to their patients, families, and peers. Fears that dentists have been reported to experience include carrying the virus to family, getting infected while treating coughing patients, and getting infected by coworkers. These fears can lead patients to undergo treatment delays, which is why it is important to develop psychological coping mechanisms and strategies to keep the practice running [110].

Countrywide shutdowns due to COVID-19 caused many to undergo a mandated quarantine. The effects of quarantine can have severe impacts on an individual. It increases the possibility of psychological and mental problems because people lack interpersonal communication, are distant from those they care about, and psychological treatment resources are severely limited [111,112]. With that in mind, it becomes essential to look for warning signs and symptoms that show someone may be suffering from mental trauma. Some may experience anxiety, depression, nervousness, anger, rumination, hopelessness, decreased concentration, insomnia, and fear [113,114]. These are some of the emotions that dentists need to be looking for, not only in their patients but also in themselves and their staff. 

Psychological distress, which often presents as fear amongst dentists, was a common experience during this pandemic. A study reported that many dentists may experience fear, anxiety, concern, sadness, and anger, but only a small percentage (8.7%) feel these intensely [10]. Another psychological effect the dental community may experience is high overload and low self-efficacy, which were associated with psychological distress amongst dentists and dental hygienists [114]. Dentists also reported fear for their professional future, such as inability to pay expenses leading them to go out of business [10,115]. The financial impact that dentists may experience has both short and long-term impacts. Some providers will go out of business, which will lead to a shortage of providers [115]. Some of this psychological distress must be ameliorated by professional improvement, such as better PPE, body temperature checks, and waiting room access. Jordanian dentists reported that they lack the minimum PPE and precaution to control infection, with 71% viewing the virus as moderately dangerous and seeing the importance of social distancing [116]. 

The psychological fear that both patients and dentists experience can ultimately influence the patients’ health outcomes. Improvements ibn oral health reduces their risk of developing the non-oral systemic disease [117]. This is why it is crucial that patients must receive care while under pandemic conditions. The American Dental Association (ADA) has created guidelines for dentists to utilize in dental emergencies. These recommendations take into account the psychological conditions of a patient. Phone triage is being used to assess the patients’ psychological and neurological functioning, resulting in patient triaging based on anxiety risk assessment. Though this cannot solely dictate if a patient has a dental emergency, it can influence the overall score [118]. 

Available data demonstrate that higher infection rates and the majority of deaths due to COVID-19 occur in assisted living homes and underserved communities due to lack of awareness, education and various psychosocial burdens [119,120,121]. As the pandemic continues, new ways to deal with dental concerns, especially in assisted living homes and underserved communities, are being implemented. As mentioned previously, telemedicine became an essential tool during COVID-19, assessing and triaging patients while also limiting contact [5,119,122]. In addition, since smokers were determined to be a high-risk group for COVID-19 complications, students and medical practitioners need to develop skills in providing smoking cessation. It is expected to see a trend towards more people wanting to quit smoking [123].

Korea is an example of implementing psychological well-being in the treatment of COVID-19. Korea has deployed mental health professionals to assist during quarantine because feelings of distress and anxiety can be exacerbated when experiencing symptoms or receiving treatment for COVID-19 [113]. Some strategies to help cope during this time are self-care and psychological flexibility. Establishing guidelines for dentists and a survey checking their mental status is an important next step for the dental community [111]. With these changing dynamics, there is a need to establish safe and secure methods for services to provide psychological counseling. In summary, high anxiety levels and significant psychosocial implications were noted among dental staff and health care workers during this pandemic. Our findings add to a growing body of data on the psychosocial impact of virus outbreaks on healthcare workers and highlight the importance of wellbeing initiatives for healthcare workers to be placed at the forefront of future pandemic crisis planning.

## 7. Potential Drugs for Fighting SARS-CoV-2 Infection and Their Interaction with Oral Health Medications

It is clinically important for oral health professionals to be aware of possible drug interactions that may occur between drugs commonly prescribed in dentistry, in order to prevent adverse reactions that may even endanger the life of a patient with COVID-19.

The ongoing pandemic caused by the SARS-CoV-2 virus has proven to be challenging in the pharmaceutical pursuit of a successful drug for treatment. New discoveries unveiling the details of the virus’s biochemical and molecular nature have helped to determine potential drugs for treatment of the COVID-19 infection. However, the need for successful clinical trials to substantiate these drugs remains. Therefore, no official FDA approved drug for the treatment of COVID-19 currently exists. There are currently several drugs being researched for treatment which will be discussed.

Perhaps the most promising drug investigated for the treatment of COVID-19 is the antiviral drug known as remdesivir [124]. Remdesivir’s overall mechanism disrupts viral replication by acting as an adenosine analog. It enters the body as a prodrug but, in its active form, can incorporate into the viral RNA via RNA-dependent RNA polymerases. This blocks the enzyme’s activity, which stops RNA synthesis in the virus [125]. The drug was noted to block the virus in vitro. It also improved the condition of an infected patient via intravenous administration [126]. Other drugs similar to remdesivir include favipiravir and ribavirin. Both of these drugs are guanine analogs that are currently approved for the treatment of other infections. There is still not enough evidence to support their use in the treatment of COVID-19 [127]. 

Lopinavir is a protease inhibitor that targets the major coronavirus protease, 3CLpro. 3CLpro is responsible for processing the polypeptide translation product from the genomic RNA into the protein components. By blocking 3CLpro the virus is unable to complete normal protein translation and cannot replicate [128]. With ritonavir as a booster, lopinavir and/or ritonavir have been shown to possess anti coronavirus activity in vitro. The efficacy of the drug has been tested in vitro and studies have shown that SARS-CoV-2 could be inhibited by lopinavir and that the drug has an acceptable EC50 [129]. However, most clinical trial studies assess the drug in combination, or in the late stages of the disease progression. Therefore, it is difficult to assess whether lopinavir/ritonavir can treat COVID-19 as a monotherapy or combined with additional drugs [129]. 

Chloroquine and hydroxychloroquine are classified as aminoquinolines and are typically used to treat malaria and autoimmune diseases such as systemic lupus erythematosus. In the treatment of SARS-CoV-2 infections, they can block the glycosylation of cell receptors of the virus. They also increases the endosomal pH required for viral fusion and have the potential to be used as broad-spectrum antiviral drugs. The use of these two drugs is included in COVID-19 treatment guidelines internationally; however, additional evidential support is needed. Clinical trials are currently being conducted to assess how safe and effective the drug is against COVID-19. One study with more than 100 patients found that chloroquine was more successful at inhibiting the exacerbation of pneumonia than the control treatment [129]. An additional study found hydroxychloroquine was even more potent than chloroquine with an EC_50_ of 0.72 μM,n possibly rendering it more effective at inhibiting the virus in vitro [130]. 

In addition, some of the medicines such as ketoconazole and erythromycin, used for dental treatment may interfere with remdesivir, lopinavir and hydroxychloroquine [131,132,133]. This in turn may worsen COVID-19 symptoms. Therefore, dental professionals should be aware of the underlying comorbidities, discuss possible drug interactions and provide an appropriate treatment regimen for COVID-19 patients visiting dental clinics. 

DMF, the only drug approved by the US Food and Drug Administration (FDA) and the European Medicines Agency (EMA) that targets the NRF2/KEAP1 axis [134], and two types of NRF2 activator were tested in advanced clinical trials, and thus can be immediately expedited to examine their therapeutic efficacy in patients with COVID-19. NRF2 activators such as sulforaphane and bardoxolone methyl are already in advanced clinical trials for other indications, providing a clear route for their testing in randomized clinical trials in patients with COVID-19.

Inhaled Nitric Oxide (iNO) is also being developed as a potential treatment for the pulmonary symptoms of COVID-19 [135]. NO is a potent vasodilator but when it is administered to a patient intravenously, it is quickly inactivated by hemoglobin. When NO is aerosolized, it can directly access lung tissue and exert its vasodilator effects on the lung’s vasculature [135]. iNO has six beneficial effects in COVID-19 patients including, anti-thrombin effects, anti-inflammatory effects, ventilation/perfusion effects, broncho-dilatory effects and microbicidal effects. This gas allows patients to have a better chance of recovery from COVID-19 while on ventilators and other ventilation aids [135]. 

Corticosteroids such as hydrocortisone and dexamethasone are also being tried out as they have shown some benefits in pneumonia and ARDS patients [136]. Corticosteroids were found to be less promising when treated for SARS and MERS [136]. In Covid-19 patients who received corticosteroids for 3–12 days mortality rate was higher than those who were not treated with corticosteroids in a meta-analysis study of about 21,350 patients [137]. Hence, there is a need to explore for an optimal duration for the use of corticosteroids in the treatment of SARS-CoV-2. 

Sex steroid hormones, especially estrogen, mount a stronger immune response in females when compared to males. As estrogen levels fall during menopause, women become more vulnerable to numerous health issues, including loss of bone mineral density which can lead to osteoporosis. Around the same time, changes in oral health are also common as teeth and gums become more susceptible to disease, which can lead to inflammation, pain, bleeding, and eventually lost or missing teeth. Estrogen therapy was shown to be effective in reducing tooth and gum diseases in postmenopausal women [138,139,140]. This protective role is lost in older adults or postmenopausal women due to decreased levels of endogenous sex hormones among COVID-19 patients [87,88]. Several clinical trials are underway using sex hormones (estrogen and progesterone) as a potential drug candidate to combat COVID-19. Jarvis et al. have discussed the combination of estrogen and progesterone to improve the immune abnormalities due to cytokine storm in COVID-19 patients [141]. Our study demonstrated that in vivo supplementation of estrogen attenuated rapid gastric emptying and restored gastric relaxation, serum NO levels, nNOSα, and normalizing Nrf2-Phase II enzymes, inflammatory response, and mitogen-activated protein kinase (MAPK) protein expression in ovariectomized diabetic rodent model [88]. We speculate that sex hormones may be helpful in suppressing COVID-19 symptoms by attenuating impaired Nrf2-NO signaling in targeted organs.

The drugs described include several possible contenders for treatment of the disease, but more evidence is necessary before an official treatment drug is endorsed. For this to occur, further in vitro, in vivo and clinical trials are warranted to determine the possible roles of the drugs in the management of COVID-19. 

## 8. New and Ongoing Research

Research for vaccines and drugs to fight COVID-19 infections has been a priority in most of the world’s institutions. Until an effective vaccine/drug is developed or discovered, reliable and efficient testing has become one of the United States most significant needs. ACE-2 expression has been found to be higher in salivary glands when compared to the lungs. Therefore, SARS-CoV-2 can be detected in saliva earlier and even before lung lesions emerge, and patients can present as asymptomatic carriers. A possible correlation between SARS-CoV-2 infections is the association with sialadenitis. The virus can cause lysis of the acinar cells in the salivary glands leaking salivary amylase into the bloodstream, leading to chronic sialadenitis. Dentists diagnosing sialadenitis may recommend that patients are tested for COVID-19 even though they might not present with the normal symptoms [27]. Salivary testing is widely used for the diagnosis of SARC-CoV-2 RNA among COVID-19 patients across the world.

Syncope or near-syncope may be a sign of COVID-19 infection [117,142]. This is still a preliminary report, only conducted in non-U.S. patients. The variation in the prevalence of tobacco use, cardiovascular disease, and dietary patterns may be confounding factors in correlating syncope with COVID-19 infection [117,142].

Vitamin D (1,25-dihydroxycholecalciferol) deficiencies have recently been linked to worse prognoses in COVID-19 infections. Vitamin D was found to increase the production of anti-inflammatory cytokines such as defensins and cathelicidins, which in turn mediate the response of the immune system to the infection. Pro-inflammatory cytokines damage lung epithelium and induce the pneumonia-like symptoms associated with a COVID-19 infection. Vitamin D deficiency may be correlated with an increased risk of “cytokine-storm” immune activity. Nutritionists recommend that people at risk of viral infections such as influenza and/or COVID-19 consider taking 10,000 IU/d of vitamin D3. After a few weeks on this regimen, Vitamin D concentrations should be increased to about 40–60 ng/mL (100–150 nmol/L). For those confirmed COVID-19 positive, a higher dose may be recommended. Research is still ongoing on the effects of Vitamin D and randomized controlled trials and large population studies should be conducted to evaluate these recommendations [143]. Vitamin D deficiencies may also be linked to why the African American population in the US may be more susceptible to the adverse risks of COVID-19. Another innovation in the fight against COVID-19 is the use of copper (Cu). Copper has three main anti-viral properties which are: (I) it damages viral envelopes and can destroy the DNA or RNA of the viruses; (II) it generates reactive oxygen species (ROS) that can kill the virus; and (III) it interferes with proteins that operate important functions for the virus. Copper supplements were suggested to be used in combination with remdesivir (RDV), N-acetylcysteine (NAC), nitric oxide (NO) and colchicine to treat COVID-19 [144]. Also, the survivability of SARS-CoV and SARS-CoV-2 on copper surfaces is much lower than on other metal surfaces. On stainless steel, SARS-CoV-2 survived for up to three days and was undetectable after four days. However, on copper SARS-CoV-2 survived for 4 h. Using copper in hospital settings and dental offices on frequently touched metal surfaces may have an increased effect on lowering the chances of surface related infections [145].

Biomarkers are critical to determining whether interventions are favorable to relieve patients from disease progression. Specific biomarkers such as cardiac and pro-inflammatory cytokines are elevated in some COVID-19 patients and in AA subjects with diabetes, hypertension and cardiovascular disease (CVD) and more recently periodontal disease [146]. Some of these biomarkers include cTn1/T, BNP, and CK-MB. Elevated cTnT was detected in COVID-19 patients with CVD, and predicted an acute myocardial injury and admission to an intensive care unit (ICU) in 4 out of 5 patients [147]. AA are diagnosed and treated for diabetes, hyper-tension and CVD at higher rates than Caucasians; however, it is yet to be determined whether biomarkers in AA are altered compared to Caucasian patients.

These and many other new strategies are being developed to combat the spread and effects of COVID-19. With continued perseverance COVID-19 will become a disease that the scientific community has well under control.

In conclusion, the SARS-CoV-2 virus has had an indelible effect globally and dentistry is not excluded. New standard protocols are implemented in dental offices worldwide and until an effective vaccine or drug is produced, much of these protocols will remain in effect long-term. A good understanding of the etiology, mechanism of infection, and epidemiology of COVID-19 will help dentists treat their patients. Knowing what co-morbidities increase the risks of fatal COVID-19 outcomes will help dentists better assess what dental procedures are worth the risks of performing on COVID-19 positive patients. In addition, staying abreast of what novel drugs are created to combat the infection is important for dentists. Patients look to dentists not only for oral health advice, though that is their main task; dentists are also expected to provide patients with holistic health advice. 

Information is generated everyday describing new signs and symptoms of COVID-19. Oral pathologies associated with COVID-19 are still being discovered and could be used in early diagnoses and/or onset of the disease. Research is still being done to look into saliva as a diagnostic tool for the virus. The antibodies associated with saliva and even the expression of certain proteins may be a correlated presence of the virus and can be used as an inexpensive and less invasive way to test and diagnose patients quickly and accurately. 

## Figures and Tables

**Figure 1 jcm-10-00156-f001:**
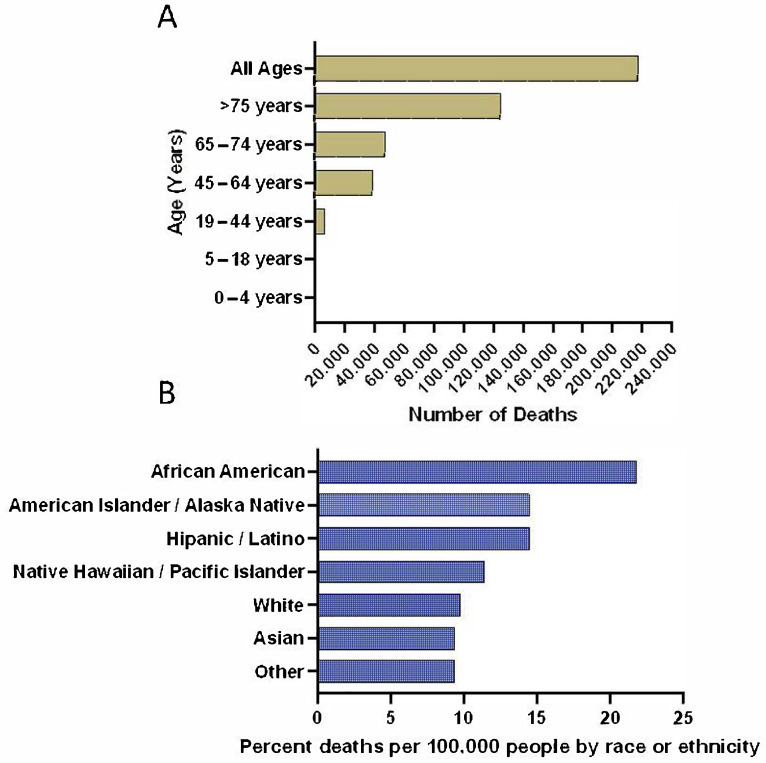
Cumulative confirmed cases of COVID-19 in United States as of November 4, 2020. The number of confirmed COVID-19 deaths by age (**A**) [12], and the number of confirmed COVID-19 deaths by race and ethnicity (**B**) [22] are presented.

**Figure 2 jcm-10-00156-f002:**
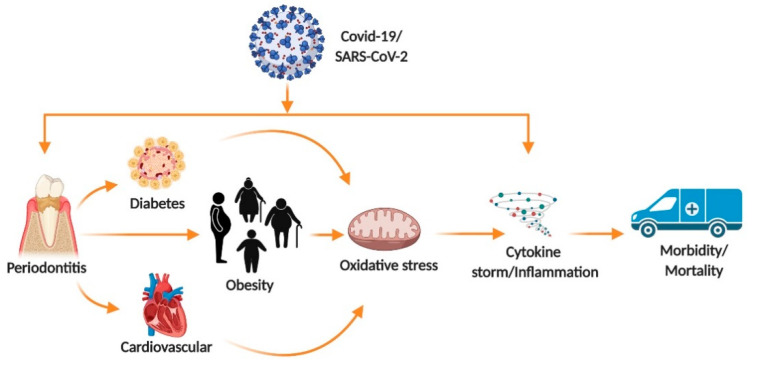
An overview of COVID-19 infection. COVID-19 infection is more pronounced in populations with comorbidities such as periodontitis, obesity, diabetes and cardiovascular disease. COVID-19 infection induces oxidative stress, triggers unregulated cytokine production (cytokine storm) and inflammation [89,90,91]. These events enhance the risk of morbidity and mortality rate in most vulnerable populations [21,23].

**Figure 3 jcm-10-00156-f003:**
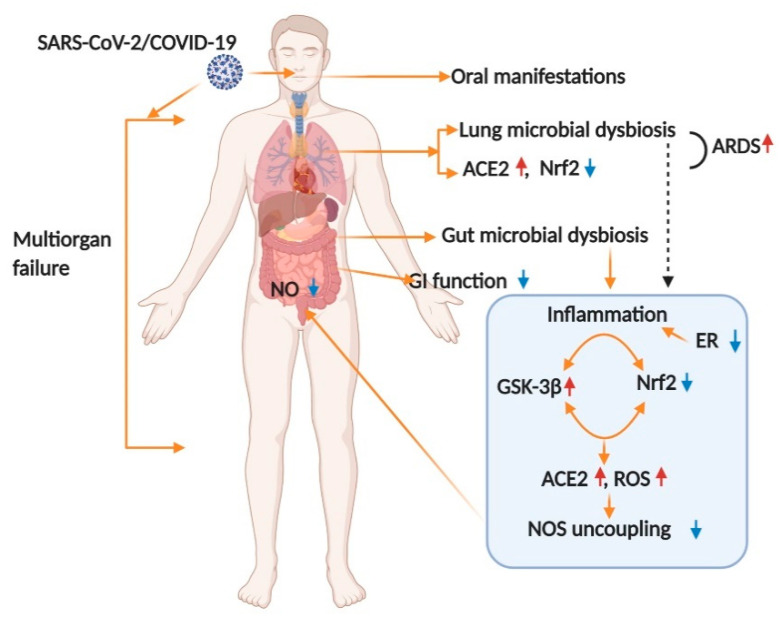
Schematic representation of the proposed mechanism involved in COVID-19-infection inducing multiple organ failure. Binding of viral protein to Angiotensin Converting Enzyme 2 (ACE-2) leads to virus entry. ACE-2 gene expression in oral tissues [46,47], lungs [48,49], vascular [71], kidney [50], stomach [51], and colon [52] has been shown to repress nuclear factor erythroid 2–related factor 2 (NRF2) [53]. A possible mechanism of ACE-2 and reactive oxygen species (ROS) activation by COVID-19 and the repressing of NRF2 executes oral manifestations, acute respiratory distress syndrome (ARDS) in lungs, inflammation and oxidative stress in multiple organs. Suppression of estrogen receptors (ER) by COVID-19 infection elevates inflammation in multiple organs. Suppression of NRF2 by COVID-19 infection reduces tetrahydrobiopterin (BH_4,_ a cofactor for nitric oxide synthase) availability, nitric oxide synthases (NOS) uncoupling, thus altering overall gastrointestinal (GI) function.

**Table 1 jcm-10-00156-t001:** Effective COVID-19 Practices for a Dental Office.

Procedures	Details	Ref
PPE, Decontamination and Sterilization Procedures	All equipment surfaces should be protected with barrier film, cleaned with hydroalcoholic disinfectants at concentrations above 60%, and then changed after every patient. It is suggested that patients use a mouth rinse of 1% hydrogen peroxide or 1% iodopovidone for 30 secs to help lower virus concentration in the mouth.	[36,103,106]
Fresh Air or Medical Grade Air Purifiers	Allow fresh air between patients either by open windows or medical-grade air purifiers.	[107]
Telephone Triage	Performing a telephone triage with patients to determine if they have symptoms or have come into contact with COVID-19 will allow dental providers the ability to screen patients. If a patient has responses that indicate, they might have come in contact inform the patent and defer treatment unless it is an emergency case.	[108]
Social Distancing	Dental offices should adhere to social distancing in the waiting room. Minimize the amount of people who have entry into operatory rooms to individual patients or a single adult for minors. All personal items should be left in the waiting room.	[105]
Temperatures of all patients, dentists, and staff are required	If the patient’s/staff/dentists’ temperatures are less than 100 °F and no COVID-19 symptoms, patients may be treated, and the dental staff and dentist may perform treatment.	[109]

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
