# Peer review of "SARS-CoV-2 Infection and Oral Health: Therapeutic Opportunities and Challenges"

_jcm, 2021, doi:10.3390/jcm10010156_

Round 1

Reviewer 1 Report

Abstract

-Please delete „ the authors hope”

Introduction

In general, the introduction is hard to read.

-Please reposit the single parts of the introduction. Write something about SARs-CoV 2 in general (first appearance, incubation time etc…..), then about the symptoms….

-Please maintain your predicates with references. For example line 60, 68,…

-line 45: The WHO wasn´t the first to identify SARS-CoV-2. Please rephrase the sentence. Please refer to the first paper identifying SARS-CoV-2. The WHO just pronounced it as pandemic.

-line 69: the description of the search strategy is not part of the introduction. Please repose it into materials and methods.

-but add the aim of the paper at the end of the introduction

Epidemiological/viral origin Data

-Please shorten the section. Maybe you merge this section with the introduction.

COVID-19 from dental perspective

- Please shorten the section. Focus on essential information. Maybe you can sum up the information in tables instead off this long text passage.

- please add some information about the research you have found to each topic.

Mechanism of infection /Potential drugs

-this are interesting sections, but they are not essential for this paper. Focus on information for oral health instead of retell overall informations.

This is an interesting paper, but it is much to long. It includes a lot of general information. The authors should focus on the oral health topic.

Author Response

We are thankful to the reviewers for their insightful critiques. We have provided point-by-point responses to reviewer critiques below.

  1. Abstract - Please delete „ the authors hope”

Response: As suggested, “the authors hope” has been deleted from the sentence (line 39, in the revised manuscript).

  1. Introduction - In general, the introduction is hard to read -Please reposit the single parts of the introduction. Write something about SARs-CoV 2 in general (first appearance, incubation time etc…..), then about the symptoms….

Response: As suggested, the authors have revised the introduction section (lines 44 – 51).

  1. Please maintain your predicates with references. For example line 60, 68,…

Response: The statements are now quoted with references (Ref # 7, 8 & 9, lines 58, 66 in the revised manuscript)

  1. line 45: The WHO wasn´t the first to identify SARS-CoV-2. Please rephrase the sentence. Please refer to the first paper identifying SARS-CoV-2. The WHO just pronounced it as pandemic.

Response: Thank you for bringing to our notice. We have rephrased the sentence (line 50, in the revised manuscript).

  1. line 69: the description of the search strategy is not part of the introduction. Please repose it into materials and methods.

Response: The authors now have removed the data collection or search strategy section from the revised manuscript for a better flow. The figure 1, associated with the data collection section also has been removed.

  1. but add the aim of the paper at the end of the introduction

Response: A brief summary/aim of the review has been added at the end of the introduction section (Line 75-78, in the revised manuscript)

  1. Epidemiological/viral origin Data - Please shorten the section. Maybe you merge this section with the introduction.

Response: As the introduction section has been revised, the authors separated the Epidemiological/viral origin Data from this section.

  1. COVID-19 from dental perspective - Please shorten the section. Focus on essential Maybe you can sum up the information in tables instead off this long text passage.

Response: The authors have condensed some of the information in COVID-19 from dental perspective section. As suggested, Table 1 is added compiling the information from the section (Line 295 & table 1 at the end of the revised manuscript).

  1. Please add some information about the research you have found to each topic.

Response: A brief summary of finding at the end of each section has been introduced as suggested.

  1. Mechanism of infection /Potential drugs - this are interesting sections, but they are not essential for this paper. Focus on information for oral health instead of retell overall informations.

Response: The mechanism of infection of the virus is very important to understand different pathways involved and the link between COVID-19 and oral and overall body health. The potential drugs introduced in this article is to highlight commonly prescribed medications by dentists that may interfere with the drugs used for COVID-19 treatment.

  1. This is an interesting paper, but it is much too long. It includes a lot of general information. The authors should focus on the oral health topic.

Response: Thank you for the positive critique. We have tried to condense the article by focusing on oral health and its importance during the pandemic or COVID-19 infection.

Reviewer 2 Report

Although the topic is a priority at this moment, the article has serious inaccuracies. There is a lot of information repeated throughout the article. The organization makes reading difficult. The order in which topics are presented doesn't make sense: it describes psychological effects first, then jumps to pre-existing conditions and outcome and then talks about the mechanisms of infection.

In the text the search terms are:Dentistry and COVID-19 (the disease) , but in figure 1a the search terms are :SARS-CoV2 (the virus) and Dentistry.

In figure 1 caption authors say:Research articles on COVID–19 (...)articles were published between 2008-2020. Of these, more than 85% were published within last five-year period. (..) Keywords used “SARS-CoV-2 and dentistry. -SARS-CoV-2 emerged in late 2019 !!!!

The statement:The World Health Organization (WHO) was the first to identify and characterize SARS- CoV-2 as the virus that caused the pneumonia of the lower respiratory tract in a patient in Wuhan,  China on December 29, 2019 is incorrect.

Some statements are not supported by bibliographic references. Others have incorrect or inappropriate references. For example it is said that :To date the age range of COVID-19 patients is between 25 and 89 years old, with most cases between 35 and 55 years. This is incorrect, it affects all ages although more severely in older ages... curiously one of the references is even from a pediatric journal journal.Recent studies showed about 45% of COVID-19 patients have oral manifestations - One of the references is a case report....

Authors say: However, only two people 75+ years died without having an underlying condition. This is incorrect also.

There are no current procedures that are in place to prevent transmission by aerosols - This isn't true

The authors speak of several controversial treatments like chloroquine, but omit steroids, which have proven to reduce mortality.

In PPE, the use of FFP2 respirators or higher in aerosol generating procedures should be suggested.

Authors waste too much time on issues that are not yet definitively proven.

For an oral health professional looking for a quick update on the topic this article falls far short

Author Response

We are thankful to the reviewers for their insightful critiques. We have provided point-by-point responses to reviewer critiques below.

  1. Although the topic is a priority at this moment, the article has serious inaccuracies. There is a lot of information repeated throughout the article. The organization makes reading difficult. The order in which topics are presented doesn't make sense: it describes psychological effects first, then jumps to pre-existing conditions and outcome and then talks about the mechanisms of infection.

Response:  The authors appreciate reviewer for an insightful suggestion.  The revised article has been re-organized with sections appearing with mechanism of action, pre-existing conditions followed by COVID-19 and dental perspectives.

  1. In the text the search terms are: Dentistry and COVID-19 (the disease), but in figure 1a the search terms are: SARS-CoV2 (the virus) and Dentistry.

Response: Data search section was removed for a better flow of the review article for the readers, hence there are no figures associated with this section in this article.

  1. In figure 1 caption authors say: Research articles on COVID–19 (...)articles were published between 2008-2020. Of these, more than 85% were published within last five-year period. (..) Keywords used “SARS-CoV-2 and dentistry. -SARS-CoV-2 emerged in late 2019 !!!!

Response: Data search section has been removed for a better flow of the review article for the readers, hence there are no figures associated with this section in this article.

  1. The statement: TheWorld Health Organization (WHO) was the first to identify and characterize SARS- CoV-2 as the virus that caused the pneumonia of the lower respiratory tract in a patient in Wuhan, China on December 29, 2019 is incorrect.

Response: The authors have revised the sentence (line 50, in the revised manuscript).

  1. Some statements are not supported by bibliographic references. Others have incorrect or inappropriate references. For example it is said that :To date the age range of COVID-19 patients is between 25 and 89years old, with most cases between 35 and 55 years. This is incorrect, it affects all ages although more severely in older ages... curiously one of the references is even from a pediatric journal. Recent studies showed about 45% of COVID-19 patients have oral manifestations - One of the references is a case report....

Response: The authors have carefully checked the references and their relevance and corrected the errors.

  1. Authors say: However, only two people 75+ years died without having an underlying condition.This is incorrect also.

Response: The authors have corrected the errors in the revised manuscript.

  1. There are no current procedures that are in place to prevent transmission by aerosols - This isn't true

Response: The authors have revised the information in the manuscript.

  1. The authors speak of several controversial treatments like chloroquine, but omit steroids, which have proven to reduce mortality.

Response: Use of steroids in the treatment of COVID-19 has been updated in the revised manuscript (Line 461 – 446, Ref 134 & 135).

  1. In PPE, the use of FFP2 respirators or higher in aerosol generating procedures should be suggested.

Response: the authors now have introduced FFP2 respirators in the revised article (Line 317)

  1. Authors waste too much time on issues that are not yet definitively proven.

Response: The review now has been extensively edited and revised. As there is so much ongoing research, it is very difficult for a definite set of treatment regime or treatment plan to protect patients against viral infection. We tried to keep the information within published research known so far.

  1. For an oral health professional looking for a quick update on the topic this article falls far short

Response: Keeping oral health professional in mind, we have carefully revised and added important information that support the importance of oral health among COVID-19 patients.